# Nature-Based Restoration Simulation for Disaster-Prone Coastal Area Using Green Infrastructure Effect

**DOI:** 10.3390/ijerph20043096

**Published:** 2023-02-10

**Authors:** Kihwan Song, Youngsun Seok, Jinhyung Chon

**Affiliations:** 1OJEong Resilience Institute, Korea University, Seoul 02841, Republic of Korea; 2Department of Environmental Science and Ecological Engineering, Korea University, Seoul 02841, Republic of Korea; 3Division of Environmental Science and Ecological Engineering, Korea University, Seoul 02841, Republic of Korea

**Keywords:** nature-based solutions, coastal resilience, restoration planning, simulation modelling

## Abstract

Floods in coastal areas are caused by a range of complex factors such as typhoons and heavy rainfall, and this issue has become increasingly serious as interference has occurred in the social-ecological system in recent years. Given the structural limitations and high maintenance costs of the existing gray infrastructure, the need for a nature-based restoration plan utilizing green infrastructure has been raised. The purpose of this study is to simulate the restoration process through the quantification of green infrastructure effects along with resilience in disaster-prone coastal areas, and to present it as nature-based restoration planning. For this purpose, first, a disaster-prone area was derived from Haeundae-gu, Busan, Republic of Korea, which was affected by typhoons. In order to simulate the runoff from typhoon “Chaba” in the target area and the effects of reducing the runoff of green infrastructure, relevant data was collected and a model constructed. Finally, the effects of the green infrastructure as applied to the disaster-prone area were quantified by means of resilience and a nature-based restoration plan was presented. As a result of this study, first, the runoff reduction effect was greatest when the maximum biotope area ratio of 30% was applied to the artificial ground. In the case of the green roof, the effect was the greatest 6 h following the typhoon passing through, and the effects of the infiltration storage facility was greater 9 h following the same. Porous pavement exhibited the lowest runoff reduction effect. In terms of resilience, it was found that the system was restored to its original state after the biotope area ratio of 20% was applied. This study is significant in that it analyzes the effects of green infrastructure based upon the concept of resilience and connects them to nature-based restoration planning. Based on this, it will be provided as an important tool for planning policy management to effectively respond to future coastal disasters.

## 1. Introduction

Throughout coastal regions, flood-related disturbances primarily ensue due to morphological characteristics resulting from these regions occurring in close proximity to the sea, and a high population density can result in significant social-ecological risk and damage [1,2]. In the case of Korea, given that many areas are located adjacent to the coast due to the geographical specificity of the peninsula, flood-induced damage due to heavy rainfall or typhoons does occur in coastal areas. Nearly 90% of the natural disasters occurring in Korea over the past decade involve floods attributed to heavy rains and typhoons [3]. As the occurrence frequency and damage scale become increasingly irregular due to the influence of climate change, it is difficult to estimate the resulting damage due to floods in coastal regions employing the same standards as those considered in the past. In addition, damage continues in terms of its social-ecological aspects as a result of artificial impacts, such as an increase in impervious areas attributed to urbanization and a lack of capacity due to structural limitations [4]. A flood occurring in the coastal region not only affects evacuation operations in its wake [5], but it also affects the ability of the system to manage the recovery process over the long term [6]. This is connected to the concept of a socio-hydrological system in which a human-centered social system and an environmental system interact and circulate for the issue of flood [7,8,9]. In other words, the hydrological process due to flooding affects the social system, and the reactions and responses in the social system are feedback to intervene in the hydrological process [10]. The aforementioned increase in impervious areas and the installation of artificial facilities such as reservoirs, drainage channels and pumps could damage ecological aspects and even impede the recovery of runoff damage due to floods [11]. In addition, due to the limitations of existing gray infrastructure which faces difficulty adapting to social and ecological changes as well as high maintenance costs [12], the need for nature-based restoration planning to compensate for coastal disasters has emerged.

In this study, nature-based restoration planning was proposed in order to respond to damage from typhoons occurring within coastal areas. This is based upon the concept of resilience, and the concept of resilience applied to coastal takes into account preparation for disasters, intermediate responses to disasters, and the process of recovery [13]. Existing gray infrastructure, such as rainwater pumping stations, is affected during recovery following flooding. In addition, artificial ground is composed of impervious areas, which increases the risk of damage due to runoff originating from the occurrence of floods to subsequent processes. Within this context, green infrastructure has been proposed as an alternative to complement the existing gray infrastructure and therefore increase resilience [14]. Green infrastructure got its start with the concept of an interconnected ecological network in green space, and was gradually integrated into the low-impact development (LID) concept or the sustainable drainage system (SuDS) to control stormwater runoff and respond to flooding [15,16]. Green infrastructure for stormwater runoff management includes green roofs, rainwater gardens, infiltration storage facilities, green spaces, and porous pavement employed throughout existing urban areas. This approach could reduce impervious areas and increase the permeability so as to delay the duration of runoff initiation [17]. Green infrastructure provides a method for increasing resilience in terms of the overall aspect of the social ecosystem, such as providing various ecosystem services, as well as the effects of reducing runoff with regard to the engineering aspect. However, the majority of green infrastructure studies have been conducted chiefly to analyze the impact of runoff through monitoring [18,19,20,21,22]. Therefore, based on the effects of these green infrastructure studies, it is necessary to investigate their use for planning to respond to coastal disasters from a long-term perspective.

To respond to disasters in coastal areas by means of utilizing green infrastructure, as mentioned above, the concept of resilience should be applied to nature-based restoration planning. The effectiveness of existing green infrastructure in reducing runoff due to flooding emerges over time within the concept of resilience. This is flood resilience in terms of engineering, which refers to the ability to return to its original state at times when the system is shocked or damaged by external disturbances [23,24]. When applied to coastal disasters, it refers to the process of restoring systems in coastal areas to their original state after damage is suffered, such as runoff due to disasters including typhoons and tsunamis [25,26]. Therefore, considering periods before and after the occurrence of coastal disasters, the process of recovering from the damage caused by flooding should be quantitatively expressed. In addition, by analyzing through simulations the runoff reduction effect, which manifests when green infrastructure is applied, and the restoration effect appearing when quantified by resilience, it may be used for nature-based restoration planning. The purpose of this study is to present a nature-based restoration plan through simulating the effects of applying green infrastructure in response to coastal disasters and changes in terms of the resilience of the system.

## 2. Materials and Methods

### 2.1. The Study Site

In this study, Haeundae-gu, a district within the city of Busan, was selected as the study site for applying nature-based restoration planning in response to coastal disasters (Figure 1). Haeundae-gu is located on the southern coast of Korea and suffers floods each year due to typhoons and heavy rains. In fact, Typhoon Maemi in 2003, Typhoon Dianmu in 2010, Typhoons Boraben and Sanba in 2012, and Typhoon Chaba in 2016 caused great damage throughout Busan, particularly in Haeundae-gu [27]. This study focuses on Typhoon Chaba, which most recently caused the greatest damage to Haeundae-gu, a district of Busan. The Haeundae-gu area is directly affected by the sea, as it is adjacent to the coast, and most residential areas are located in the lowlands [11], making it generally more sensitive to flood damage. As the ratio of impervious areas is high, due to the urbanization of densely populated residential areas within the coastal area, the damage is more severe due to the lack of green infrastructure which could act as a buffer mitigating floods [28].

In this study, using data from the study by Song et al. [29], the effect of green infrastructure on floods in Haeundae-gu and a plan through resilience quantification were presented. The result of the suitable green infrastructure area analyzed for Haeundae-gu in the previous study [29] was also utilized in this study. Accordingly, in the subsequent results, some differing results were derived depending upon the factors or viewpoints to be presented in this study. In addition, the drawing of conclusions and the presentation of their interpretations was sought.

### 2.2. Data Collection and Model Construction

This study collected data for the express purpose of simulating flood damage in Haeundae-gu, a district of Busan, due to Typhoon Chaba, in 2016 (Table 1). Typhoon Chaba hit Busan on 5 October 2016, resulting in a daily maximum of 128.5 mm of precipitation and an hourly maximum of 38.3 mm [30]. For the data related to the area of Haeundae-gu, the previous study of [25] was used, and in this study, a total of 283,367 m^2^ was derived for the applicable area of green infrastructure in Haeundae-gu. Following [31,32], total areas to which the green infrastructure can be applied are currently being classified as public areas (33,531 m^2^), private areas (83,684 m^2^), green areas (46,472 m^2^), transportation (94,735 m^2^) and industrial areas (24,945 m^2^).

In addition, this study limited the sorts of green infrastructure applicable to artificial ground in Haeundae-gu to three types. In reality, the green infrastructure in coastal areas features a variety of natural infrastructure such as salt marshes, wetlands, dunes, mangroves, and coral reefs. However, green infrastructure in this study focused on green roof, infiltration storage facility, and porous pavement that can be applied to urban artificial ground adjacent to the coast, rather than completely natural elements such as natural (or eco-) infrastructure [29]. In data related to green infrastructure, the values displayed in Table 1 were applied to the model and analyzed, focusing on the effects of reducing runoff due to flooding. In the case of green roofs, the reduction efficiency was 97% when the sum of precipitation was less than 32 mm, and the reduction efficiency was 70% when it exceeded 32 mm [33]. In the case of an infiltration storage facility, Table 1 shows the reduction efficiency of runoff according to total rainfall based on a study which calculated a regression equation from data covering total rainfall and reduction. In the case of porous pavement, a minimum of 15.4 to a maximum of 37.1% of the runoff reduction effect was applied [34]. Based on the data collected on site characteristics, area according to land cover type, and green infrastructure in Haeundae-gu, a district of Busan, a model for quantifying the effects and resilience of green infrastructure was built and used for simulation (Figure 2). Stella Architect 2.1.1 was used for construction and simulation of the model based on the collected data.

**Table 1 ijerph-20-03096-t001:** Data collection in Haeundae-gu, a district of Busan for the simulation model.

Characteristics	Parameters	Values	Units	Source
Site	Precipitation for Typhoon “Chaba”	38.3	mm/hour	[30]
	Duration of precipitation	12	hours	[30]
	Total (Impervious+ Pervious) area	283,367	m^2^	[29]
	Impervious area	236,895	m^2^	[29]
	Pervious area	46,472	m^2^	[29]
	Runoff amount	30,441.01	m^3^	[29,30]
Land cover	Public area (public facility and culture, sports, and recreation areas)	33,531	m^2^	[29]
	Private area (commercial and residential areas)	83,684	m^2^	[29]
	Transportation area	94,735	m^2^	[29]
	Industrial area	24,945	m^2^	[29]
	Green area	46,472	m^2^	[29]
Green infrastructure	Green roof rate	97 (≤32 mm), 70 (≥32 mm)	%	[33]
	Infiltration storage facility rate	0.0921 × accumulated rainfall + 89.606	%	[35]
	Porous pavement rate	15.4~37.1	%	[34]

### 2.3. Green Infrastructure Effect and Resilience Simulation

Based on the structure of the model built during the previous step, some scenarios were presented to analyze the effects of green infrastructure on floods. First, the change in flood damage as generated by typhoons and the resilience quantifications were analyzed according to the area where the green infrastructure in question was applied. During the process, scenarios were presented by increasing the application of green infrastructure to 10%, 20%, and 30% levels according to the biotope area ratio of Korea. The biotope area ratio quantifies the area of green infrastructure which may respond to climate change and the recovery of natural ecological functions outside of the artificial area [36]. In the course this study, we attempted to analyze the effect by applying green infrastructure step by step within the maximum area of 30% of the biotope area ratio. Second, the change in flood damage and the quantification of resilience were both analyzed by means of applying green infrastructure differently for each sort of land cover. As shown in Table 1, the artificial ground at the site can be classified into public areas, private areas, transportation areas, and industrial areas. Public areas consist of public facilities and culture, sports and recreation areas, etc. Thus, three types of green infrastructure can be applied. Private areas consist of commercial areas and residential areas, so all three types of green infrastructure can be applied there. However, since the transportation area is chiefly a road area and the industrial area mainly contains factories, it is relatively difficult to apply green roofs. Therefore, a scenario was developed by applying an infiltration storage facility and porous pavement.

In the process of quantifying flood damage caused by typhoons and the effects of reducing them through green infrastructure as resilience, the concept of engineering resilience was used. This is expressed as an index and compared through normalization with resilience based on the simulation results derived in the previous step. The process of recognizing flood damage as a function of the system and normalizing it into resilience referred to existing engineering resilience studies [11,37]. The values normalized over time were able to compare the effects of the restoration process in terms of resilience for each scenario through the four elements of engineering resilience: robustness, redundancy, resourcefulness, and rapidity. The 4R concept comprises components used during the dynamic quantification process in terms of engineering resilience, and the 4R can largely be divided into periods before and after the point at which the system is recovered following damage occurrence [38]. After the system is damaged, robustness and redundancy interfere in the process prior to recovery. Robustness refers to the ability of the system to withstand external disturbance or damage while maintaining its function, and redundancy refers to a resource which can be replaced in the process of system damage [38]. Considering the subsequent process of system recovery, resourcefulness and rapidity can give rise to interference. Resourcefulness refers to the ability to move resources required for recovery after a system has been compromised, and rapidity refers to the ability to contain losses due to impairment and achieve goals for recovery [39]. The four properties of each resilience are able to be expressed as slope, time, and *y*-axis values on the graph, as shown in Figure 3. Since these values are produced through indexation, rather than each value itself having a standalone meaning, it is a measure of how large or small it is when compared for each scenario.

## 3. Results and Discussion

### 3.1. Green Infrastructure Effect Simulation

#### 3.1.1. Green Infrastructure Application Area Scenario

Scenarios were classified according to the ratio of green infrastructure (green roof, infiltration storage facility, porous pavement) applied to the entire area of the study site. Under the basic scenario, green infrastructure was not applied, and under scenarios 1, 2, and 3, the green infrastructure application rate was adjusted to 10%, 20%, and 30%, respectively. The resulting changes are shown in Figure 4, and the values are summarized in Appendix A below. When Typhoon Chaba made landfall, it rained for approximately 12 h. The peak discharge occurred twice, at approximately 6 and 9 h following the onset of rainfall. Six hours after the beginning of the typhoon, when the first peak runoff occurred, the runoff amount declined by 7.6%, 15.3%, and 22.9% for each scenario. Nine hours after the start of the typhoon, when the second peak runoff occurred, the runoff decreased by 6.9%, 13.9%, and 20.8% for each scenario.

#### 3.1.2. Green Infrastructure Application Scenarios by Land Cover Type

As a result of the previous step (Section 3.1.1) depending on the area in which the green infrastructure can be applied, the effect of reducing the runoff was most noticeable when the ratio of applying green infrastructure to the study site stood at 30%. Accordingly, a scenario was established to apply the type of green infrastructure by region (public, private, industrial, transportation) under the premise that the application rate of green infrastructure in the entire target area reached 30%. A scenario involving the application of green roofs, infiltration storage facilities, and porous pavement in public and private areas was established. In the industrial and transportation areas, infiltration storage facilities and porous pavement were applied. This was performed via analysis of the industrial and transportation areas in Haeundae-gu, and given that it is difficult to employ green roofs, only the other two types were applied.

Four scenarios, including the basic scenario, were applied to the public area, and the runoff reduction effect was analyzed. In the case of the basic scenario, green infrastructure was not applied. In addition, green roof was applied in Scenario 1, infiltration storage facility applied in Scenario 2, and porous pavement applied in Scenario 3, at a rate of 30%. The effects of runoff reduction under each scenario are shown in Figure 5 and Appendix A. Two peak discharges occurred at the 6 and 9 h. When the first peak discharge occurred, the runoff amount declined by 38.8%, 35.1%, and 15.3% in scenarios 1, 2, and 3 compared to the basic scenario, respectively. When the second peak runoff occurred after 9 h, the runoff amount fell by 28%, 33.2%, and 15.3% in scenarios 1, 2, and 3 compared to the basic scenario, respectively. In the beginning, the runoff reduction effect of green roof according to Scenario 1 was the highest (38.8%), but it can be seen that the effect of the infiltration storage facility according to Scenario 2 increases over time (33.2%). The application of porous pavement according to Scenario 3 was effective compared to the basic scenario, but the overall effect was low as compared to other green infrastructures.

In the case of the private area, as in the public area, the three types of green infrastructure were applied. The application rates stood at 30% for green roofs (Scenario 1), infiltration storage facilities (Scenario 2), and porous pavement (Scenario 3).

When the first peak discharge occurred after 6 h, the runoff amount reductions were 30.6%, 27.7%, and 12.1% in scenarios 1, 2, and 3, respectively, compared to the basic scenario. Additionally, in the case of the second peak discharge after 9 h, the runoff amount was reduced by 22.1% in Scenario 1, 26.2% in Scenario 2, and 12.1% in Scenario 3. Specific details are presented in Figure 6 and Appendix A.

In the industrial and transportation areas, Scenarios 1 and 2 were presented, in which 30% of the infiltration storage facility and porous pavement were applied, excluding the green roof. In the case of the basic scenario, as in the previous public and private areas, the green infrastructure was not applied. In Scenario 1, where the infiltration storage facility was applied when the first peak discharge occurred following a period of 6 h, about 29.2% of runoff reduction occurred as compared to the basic scenario. In Scenario 2, during which porous pavement was applied, the runoff amount was reduced by about 12.7% compared to the basic scenario. As the second peak discharge occurred after 6 h, in Scenario 1, runoff amount was reduced by about 27.7% compared to the basic scenario, and in Scenario 2, the runoff amount was reduced by 12.7%, the same as the first peak runoff. Details on these findings are shown in Figure 7 and Appendix A.

In the transportation area, the rate of runoff amount reduction was the same as that observed in the industrial area. Since the transportation area appears to be a higher area than the industrial area, the amount of runoff reduction itself appears much greater. However, since the amount of runoff reduced per area is the same, the reduction rate of runoff amount is the same regardless of the type. Although there was a difference in area by type of land cover, it is because the ratio of reduction according to infiltration storage facility and porous pavement is the same. Specific details are shown in Figure 8 and Appendix A.

### 3.2. Resilience Quantification Simulation

#### 3.2.1. Change in Resilience by Green Infrastructure Application Area

In this step, the resilience value of green infrastructure was analyzed. Under the basic scenario, green infrastructure was not applied, and accordingly, the resilience was reduced to 0.1181 after 13 h. In addition, redundancy, which decreased for 13 h after the start of rainfall, was expressed as a function of time (−0.0894t + 1.2528). In the case of resourcefulness and rapidity, values were not derived given that the resilience could not be restored to the original state. The change in resilience according to the basic scenario is shown by the black solid line in Figure 9.

The change in resilience according to scenario 1 is displayed as the red long-dashed line in the graph below (Figure 9). Following the typhoon occurred, the robustness value stood at 0.1855, which is higher than in the basic scenario. The slope of redundancy (−0.0826t + 1.234) is larger than the basic scenario, which means that the resilience decreased slowly after the typhoon occurred in scenario 1. However, as in the basic scenario, the value of resourcefulness and rapidity was not derived because the resilience did not recover to its original state for 48 h after the typhoon occurred.

Under Scenario 2, where 20% of the green infrastructure is applied, the change in resilience is shown as a blue dashed–single-dotted line in Figure 9. The values of robustness and redundancy were larger than those in the basic scenario and Scenario 1 (Table 2). The slope in the process of restoring to its original state after 13 h, when the resilience was minimal, was expressed as resourcefulness (0.0234t − 0.1265). In addition, the rapidity, which is the time for the value of resilience to be restored to its original state following the occurrence of a typhoon, stood at 47 h.

For Scenario 3, where the green infrastructure is applied at a maximum of 30%, the change in resilience is expressed as a green dash–double-dotted line (Figure 9). In the case of robustness and redundancy, it was the largest value in all scenarios. The slope representing resourcefulness appears smaller than is the case in Scenario 2, indicating that the speed at which resilience recovers is slower. In the case of rapidity, it is 46 h, which is 1 h shorter than in Scenario 2. Comparing Scenarios 2 and 3 comprehensively, it can be seen that Scenario 3 is more effective than Scenario 2 in terms of robustness, redundancy, and rapidity in terms of resilience. However, in the case of resourcefulness, Scenario 3 recovers more slowly than Scenario 2. This is due to the applied green infrastructure affecting the delay in the occurrence of runoff after the typhoon, but the facility that discharges the runoff rainwater to the sea or river, such as the rainwater pumping station, has a greater impact on the recovery process. A detailed discussion of this analysis can be found in the discussion section that follows.

#### 3.2.2. Change in Resilience by Land Cover Type

In the second step, the change in resilience was analyzed when the green infrastructure was applied differently according to the land cover type (public, private, industrial, transportation areas) of the target area. In the process, the scenarios of each land cover type and the analysis results of the runoff reduction in Section 3.1.2 were utilized. In the case of the public area, the change in resilience is shown in Figure 10 and Table 3. The robustness value was the highest in Scenario 2, where only the infiltration storage facility was applied, followed by Scenario 1 (green roof only), Scenario 3 (porous pavement only), and basic scenario (no green infrastructure) in that order. In terms of robustness, the resilience value was the lowest after 11 h under all scenarios. Accordingly, the redundancy also captured the change in resilience value as the slope from the start of rainfall to 11 h. Accordingly, resilience appeared high in the order of Scenario 2, Scenario 1, Scenario 3, and the basic scenario. Resourcefulness is the rate of recovery of resilience after 11 h, and the basic scenario and Scenario 3 were larger than Scenario 1 and Scenario 2. It is observed that there is a difference due to the value of robustness and rainwater pumping station. Rapidity showed the shortest recovery time at 29 h following the resilience decreased in Scenarios 1 and 2. Next, Scenario 3 appeared in the order of 31 h, and for the basic scenario it was 33 h. As a result, the green roof-applied Scenario 1 and infiltration storage facility-applied Scenario 2 recovered the most quickly from typhoon damage, and the effect of porous pavement-applied Scenario 3 was the lowest. To sum up the properties of the resilience, the resilience of Scenario 2 with the infiltration storage facility was the highest overall. Scenario 1, to which the green roof was applied, had some properties such as rapidity identical to those of scenario 2, but other properties were relatively lower than those of Scenario 2. Scenario 3 with porous pavement showed the lowest resilience as compared to other scenarios and showed a large difference in overall value of resilience properties.

In the private area, the order of the each scenario for resilience was similar to that of the public area, but the specific values were different from those in the public area. The lowest resilience occurred 13 h after the occurrence of typhoon. As a result, the robustness value was indicated by the resilience value after 13 h, and the redundancy was indicated by the slope of the resilience decreased by 13 h. In the case of resourcefulness, the resilience did not recover to its original level after 48 h in the basic scenario. Under scenarios 1 and 2, the recovery period lasted in a range of 13 to 44 h, while under scenario 3, the slope persisted throughout the entire period up to 47 h. Accordingly, the value of rapidity also differed. Details of the results are shown in Figure 11 and Table 4.

In the industrial area, the resilience of Scenario 1, in which only the infiltration storage facility was applied, exhibited a higher effect than the basic scenario and Scenario 2. The robustness is the resilience value 13 h after the occurrence of the typhoon, and the redundancy is the slope of the resilience that has decreased over time until 13 h. In Scenario 1, robustness was the largest, and the slope of redundancy was the smallest. The resilience observed in Scenario 2 with porous pavement was lower than in Scenario 1 but higher than in basic scenario. Resourcefulness showed a slope from 13 h to 32 h in the basic scenario, from 13 h to 28 h in Scenario 1, and from 13 h to 30 h in Scenario 2. Accordingly, the rapidity was shortened in the order of Scenario 1 (28 h), Scenario 2 (30 h), and the basic scenario (32 h). The specific values for each resilience characteristic are shown in Figure 12 and Table 5 below.

Finally, although the transportation area differed from the industrial area in numerical terms, the tendency of resilience change by scenario was similar. The analysis results for each resilience property are shown in Figure 13 and Table 6 below. In the case of the basic scenario, the values of resourcefulness and rapidity were not derived, given that the original state was not restored even following 48 h after the typhoon occurred. Although the values of robustness and redundancy are different, the order of high resilience was the same as the results of the industrial area. Resourcefulness was shown as a resilience slope from 11 h to 44 h in Scenario 1 and from 11 h to 48 h in Scenario 2. Accordingly, the rapidity was 44 h and 48 h in Scenarios 1 and 2, respectively.

### 3.3. Discussion: Nature-Based Restoration Planning

In this step, nature-based restoration planning was proposed based upon the results of the stormwater runoff reduction effect and the properties according to the change in resilience. First, we proposed a plan in order to enhance resilience by means of improving the four properties of resilience: robustness, redundancy, resourcefulness, and rapidity. In the previous results (Section 3.1.1), when green infrastructure is applied at the maximum value (30%) of the biotope area ratio, the four properties of resilience are improved overall [37,38]. Green infrastructure (green roof, infiltration storage facility, porous pavement) in this study, which could replace artificial surfaces in urban areas, could reduce and delay the amount of runoff flowing into facilities treating local rainfall, such as rainwater pumping stations. Therefore, green infrastructure reducing the peak discharge and delaying runoff could be approached in terms of robustness and redundancy prior to recovery. In addition, in order to increase robustness, it is necessary to improve the runoff reduction effect of the green infrastructure which is to be applied. The higher the reduction effect of the runoff, the more the resilience can be improved because it can be defended against flood damage such as typhoons [38]. Since redundancy is an alternative resource in the process of flood damage such as typhoons and indicates how slowly the resilience decreases [38], it is important to apply various green infrastructures. Redundancy potentially provides ‘insurance’ for response to typhoons, by allowing some green infrastructure to compensate for the loss or failure of others [40,41]. Therefore, even if runoff reduction effect of the infiltration storage facility is the highest among various scenarios, collectively applying a single green infrastructure may not be effective in terms of redundancy. By land cover type, plans to apply green roofs to building areas, infiltration facilities to other artificial grounds, and porous pavements to spaces such as parking lots or road pavements are proposed. If resilience to floods was recognized in terms of resourcefulness, it could be regarded as a recovery process after the occurrence of system damage due to floods. From a physical perspective, this represents the presence or capacity of a rainwater pumping station. When damage occurs as a result of excess runoff, rainwater pumping stations can treat and discharge water into the sea, so these facilities could be considered an influencing factor on resourcefulness. Most studies related to gray infrastructure, such as existing rainwater pumping stations, focus not merely on the evaluation of the flood reduction function of the facility itself [42], but also on the goal and supplementation of the facility in terms of operation [43]. Therefore, as mentioned in the introduction, the application of green infrastructure can be presented as a plan for increasing resourcefulness by supplementing the existing gray infrastructure. In addition, as previously analyzed, green infrastructure could also indirectly affect resourcefulness due to the fact that existing elementary schools or small parks could be used as spaces to evacuate or mobilize resources after flood damage [44]. When considering green infrastructure and existing facilities as a whole, the above planning process and application considerations could improve resilience by increasing rapidity. In this study, the effect of the existing rainwater pumping station on the concept of resilience and the result integrated with green infrastructure are presented as nature-based restoration planning.

Second, when green infrastructure is applied, it is chiefly used for green roofs and infiltration storage facilities. In all areas (public, private, industrial, and transportation), the runoff reduction effect and resilience in scenarios where porous pavement was applied were lower than those of other green infrastructures (green roof, infiltration storage facility). Since the effect of porous pavement is lowest compared to other green infrastructures, even when planning, green roofs and infiltration storage facilities should be applied as frequently as possible, and porous pavement should be applied to other artificial ground areas. If a plan is presented for each land cover, in the case of roof greening, it can be mainly applied to public and private areas, but it is difficult in practice to apply to transportation and industrial fields. In contrast, infiltration storage facilities can be applied across all areas in various forms (bioretention facilities, trenches, etc.), and the same is true for porous pavement [36]. Therefore, when each element is applied in impervious areas identified as suitable areas for green infrastructure application, it is necessary to determine which facilities are actually effective. The application of green infrastructure can provide various methods for increasing resilience in terms of the social-ecological system and runoff reduction and delay effects, as mentioned above in regard to the four resilience characteristics.

To this end, the target area of this study, Haeundae-gu, Busan, was specifically explored. Artificial ground in the Haeundae-gu area is classified into the Bansong district, Banyeo district, Songjeong district, U district, and Jaesong district. In the case of the Bansong district, most of the areas are private, and the balance comprises transportation and industrial areas. Thus, to improve resilience in the Bansong district, green roofs must be applied to the available buildings as much as possible, up to a rate of 30%. Infiltration storage facilities should then be connected to the green roofs. Porous pavement should be applied to the spaces where green roofs or infiltration storage facilities cannot be applied in order to bring the total green infrastructure rate up 30%. In the case of the Banyeo district, the proportions of private areas and industrial areas are similar. Green roofs should preferentially be applied in private areas, and infiltration storage facilities should preferentially be applied in industrial areas. The sites in the Songjeong district primarily consist of transportation and private areas. Infiltration storage facilities should be primarily applied in the transportation area given that it is larger than the private area, and green roofs should also be applied within private areas. The U district is overall composed of transportation areas, followed by public and private areas. Infiltration storage facilities should be applied in transportation areas, and green roofs should be applied equally in public and private areas. Porous pavement should be applied in additional areas, wherein the proportion of the total green infrastructure does not exceed 30%. In the case of the Jaesong district, green roofs should be preferentially applied, and infiltration storage facilities and porous pavement should be applied in additional areas in light of the fact that private areas are predominate. In summary, the application of green infrastructure mainly depends on the proportions of public, private, transportation, and industrial areas in the target area. Therefore, the plans to utilize green infrastructure to improve the resilience will differ according to the characteristics of the site.

Nature-based restoration planning which can increase resilience in response to coastal disasters should suggest a process for applying green infrastructure to artificial ground and combining it with existing gray infrastructure so as to restore it over time. In the case of replacing artificial ground, it is more important to apply green infrastructure up to the maximum value according to the biotope area ratio in Korea when addressing damage caused by floods. In the process of applying this green infrastructure, it is applied as frequently as possible to facilities which can support green roofs in public and private areas. In connection with the green roof, infiltration storage facilities such as infiltration detention ponds, rainwater gardens, and bioswales are utilized, and porous pavement is applied to spaces such as parking lots. Since it is difficult to practically apply a green roof in transportation and industrial areas, the combination of an infiltration storage facility and porous pavement contributes to the restoration of damage caused by floods. Since conventional coastal disaster planning focuses on the coastal region itself, the green infrastructure to be dealt with focuses on breakwaters, coral reefs, mangroves, and sand dunes [45]. Accordingly, even when presenting a plan for coastal disasters, the effect on the coastal green infrastructure was shown and synthesized from a macroscopic point of view [46]. In addition, most of the studies focused on the effects of the experimental aspects of each green infrastructure element and could not be connected to the plan [33,34,35]. Therefore, it is important for green infrastructure to simulate the restoration process for damage based on the results of analyzing the effects of flooding caused by Typhoon Chaba as well as connect it to planning.

## 4. Conclusions

This study is meaningful in that it analyzes the application and effect of green infrastructure in coastal areas in terms of resilience and connects it to nature-based restoration planning. Based on the case of Typhoon Chaba, which affected Haeundae-gu, a district of Busan, and caused great damage, the effect of applying green infrastructure was analyzed and quantified as resilience. In the process, scenarios were constructed according to the applicable area and type of land cover. The primary results of this study are as follows. First, when the maximum biotope area ratio (30%) was applied, the flood reduction effect was the greatest, and the resilience was the highest. Second, when applied to different land cover types (public, private, transportation, and industrial areas) for each green infrastructure, the infiltration storage facility was the most effective, followed by the green roof. Porous pavement exhibited the lowest effect. In the initial peak discharge, the effect of the green roof appears to be higher than that of the infiltration storage facility, but as time passes, the effect of the infiltration storage facility becomes increasingly significant. Third, based on these results, nature-based restoration planning was duly presented.

To apply the planning presented within this study to areas which are actually damaged by coastal disasters, empirical research should be conducted during the future. In this study, each type of green infrastructure was applied collectively according to land cover within a regional scope, and the effect was analyzed in terms of resilience. However, for these results to be applied to actual areas and prove the effect, a test bed must be set inside a more specific unit and range, and green infrastructure (green roof, infiltration storage facility, porous pavement) must be applied. Based on this, it is necessary to prove the validity in connection with the simulation as seen in this study. The results of this study can be presented as specific evidence necessary for policies or studies which focus on the individual functions of existing green infrastructure or present macroscopic plans focusing on coastal infrastructure (breakwater, sand dune, salt marsh, etc.). The nature-based restoration planning proposed based on this scientific process can be employed as an essential plan in the policy for management in terms of resilience against the disturbance of coastal disasters.

## Figures and Tables

**Figure 1 ijerph-20-03096-f001:**
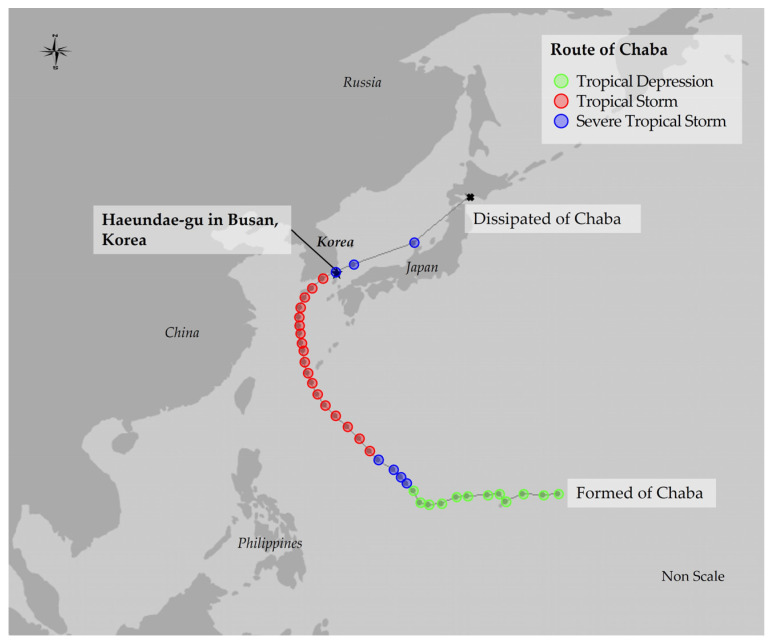
Study site and Chaba’s track.

**Figure 2 ijerph-20-03096-f002:**
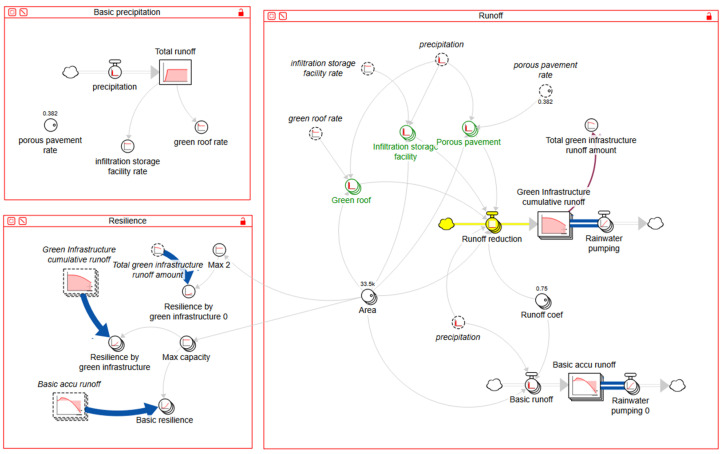
Model structure.

**Figure 3 ijerph-20-03096-f003:**
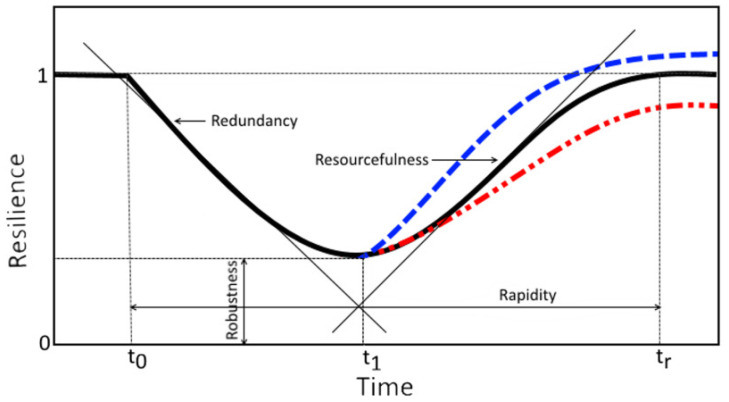
Four properties of resilience (i. Black solid line: resilience returns to pre-disturbance level; ii. Blue dashed line: resilience exceeds pre-disturbance level; iii. Red dashed–dotted line: resilience does not return to pre-disturbance level).

**Figure 4 ijerph-20-03096-f004:**
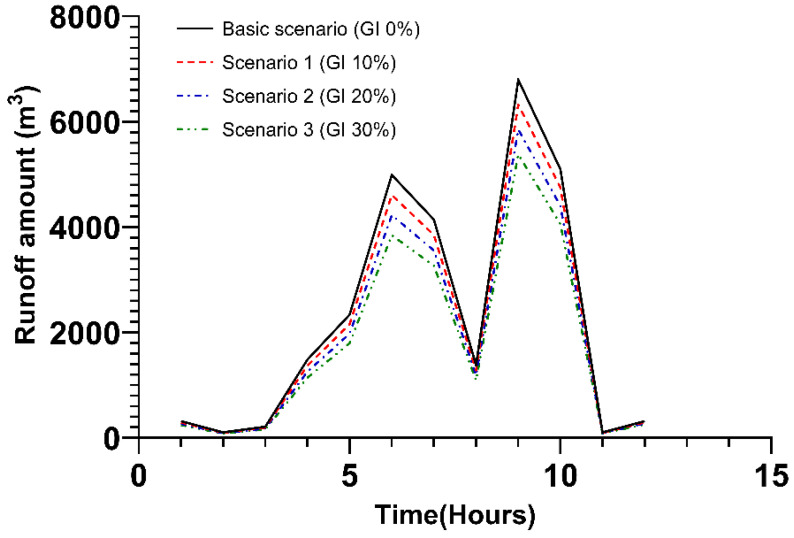
Runoff amount under the application of green infrastructure in the total area. GI: Green Infrastructure.

**Figure 5 ijerph-20-03096-f005:**
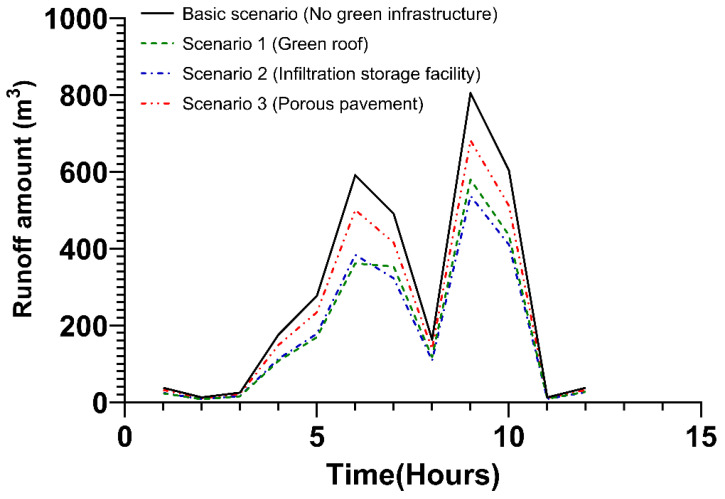
Runoff amount by application of green infrastructure in the public area.

**Figure 6 ijerph-20-03096-f006:**
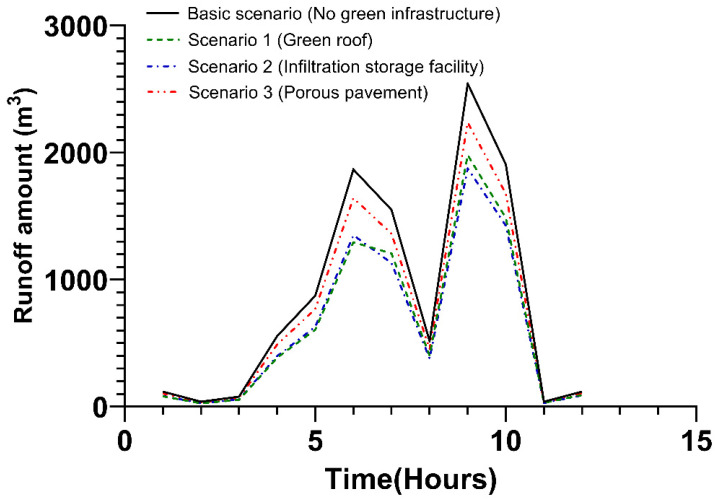
Runoff amount due to the application of green infrastructure in the private area.

**Figure 7 ijerph-20-03096-f007:**
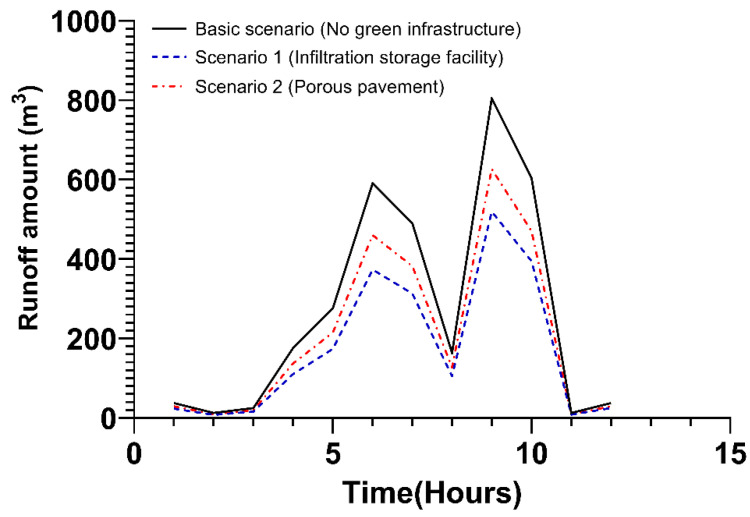
Runoff amount due to the application of green infrastructure in the industrial area.

**Figure 8 ijerph-20-03096-f008:**
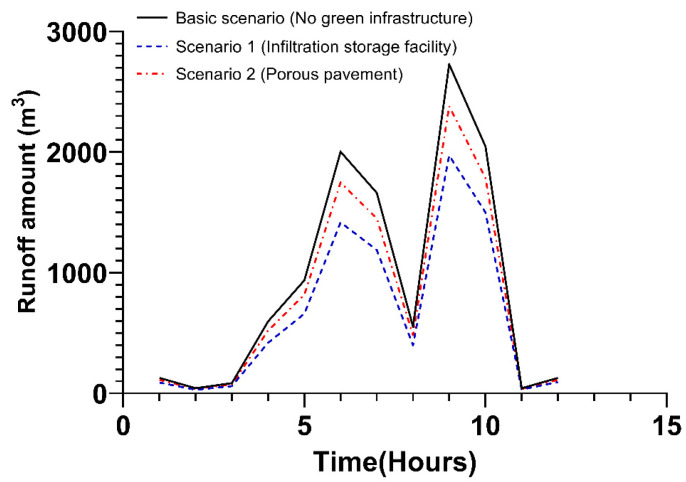
Runoff amount via the application of green infrastructure in the transportation area.

**Figure 9 ijerph-20-03096-f009:**
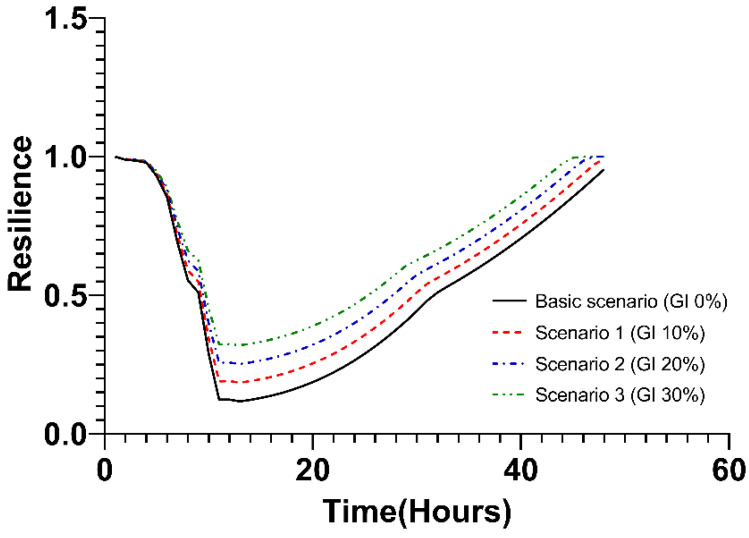
Changes in resilience values over time. GI: Green Infrastructure.

**Figure 10 ijerph-20-03096-f010:**
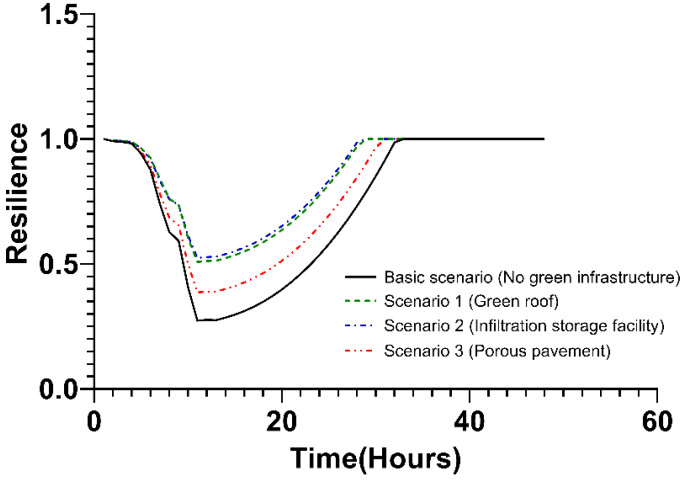
Changes in resilience values over time in the public areas.

**Figure 11 ijerph-20-03096-f011:**
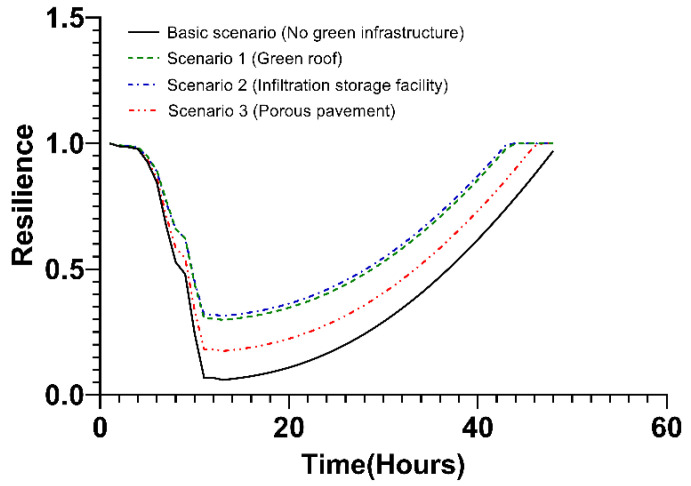
Changes in resilience values over time in the private areas.

**Figure 12 ijerph-20-03096-f012:**
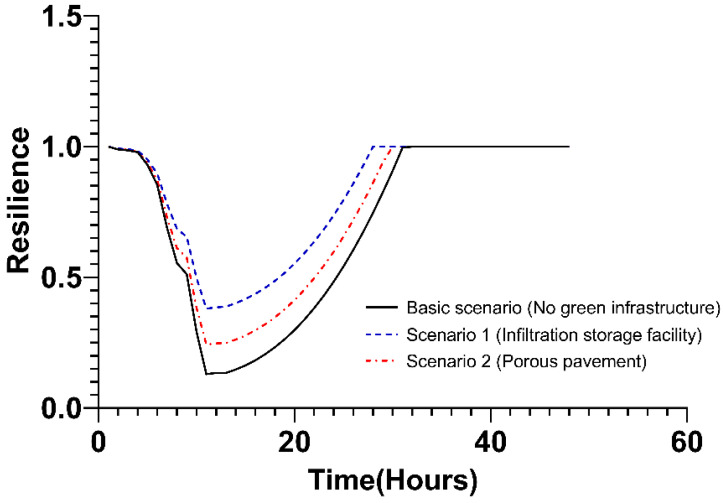
Changes in resilience values over time in the industrial area.

**Figure 13 ijerph-20-03096-f013:**
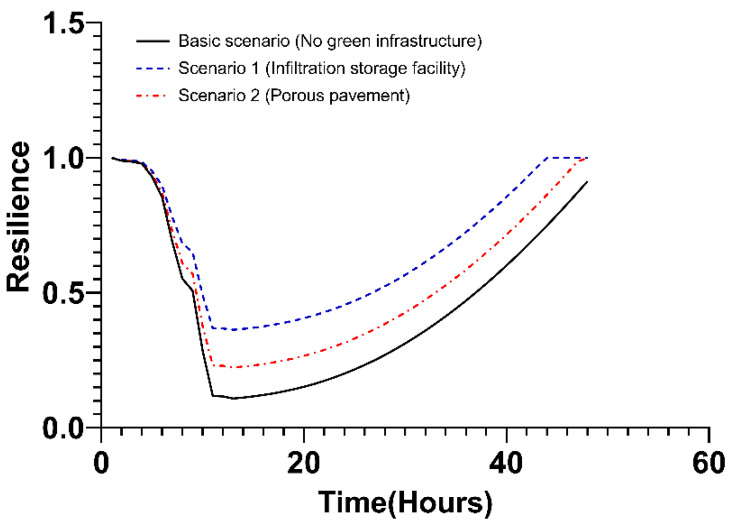
Changes in resilience values over time in the transportation area.

**Table 2 ijerph-20-03096-t002:** The 4R value according to the green infrastructure area.

Scenario	Resilience			
	Robustness	Redundancy	Resourcefulness	Rapidity (h)
Basic Scenario (GI: 0%)	0.1181	−0.0894t + 1.2528	-	-
Scenario 1 (GI: 10%)	0.1855	−0.0826t + 1.234	-	-
Scenario 2 (GI: 20%)	0.2529	−0.0758t + 1.2153	0.0234t − 0.1265	47
Scenario 3 (GI: 30%)	0.3203	−0.0689t + 1.1965	0.0223t − 0.0367	46

**Table 3 ijerph-20-03096-t003:** The 4R value in the public area.

Scenario	Resilience			
	Robustness	Redundancy	Resourcefulness	Rapidity (h)
Basic scenario (No green infrastructure)	0.2735	−0.0178t + 0.034	0.0032t − 0.0344	33
Scenario 1 (green roofs)	0.5083	−0.0128t + 0.0285	0.003t − 0.0328	29
Scenario 2 (infiltration storage facilities)	0.5239	−0.0118t + 0.0233	0.003t − 0.0305	29
Scenario 3 (porous pavement)	0.3864	−0.015t + 0.0286	0.0032t − 0.0347	31

**Table 4 ijerph-20-03096-t004:** The 4R value in the private area.

Scenario	Resilience			
	Robustness	Redundancy	Resourcefulness	Rapidity (h)
Basic scenario (No green infrastructure)	0.0603	−0.0104t − 0.0227	–	–
Scenario 1 (green roofs)	0.2983	−0.0084t − 0.0119	0.0014t − 0.0158	44
Scenario 2 (infiltration storage facilities)	0.3143	−0.0078t − 0.0154	0.0014t − 0.0152	44
Scenario 3 (porous pavement)	0.1749	−0.009t − 0.0201	0.0014t − 0.0152	47

**Table 5 ijerph-20-03096-t005:** The 4R value in the industrial area.

Scenario	Resilience			
	Robustness	Redundancy	Resourcefulness	Rapidity (h)
Basic scenario (No green infrastructure)	0.1300	−0.0214t + 0.0435	0.004t − 0.0435	32
Scenario 1 (infiltration storage facilities)	0.3804	−0.0154t + 0.0298	0.0044t − 0.0477	28
Scenario 2 (porous pavement)	0.2428	−0.0186t + 0.0351	0.0042t − 0.0474	30

**Table 6 ijerph-20-03096-t006:** The 4R value in the transportation area.

Scenario	Resilience			
	Robustness	Redundancy	Resourcefulness	Rapidity (h)
Basic scenario (No green infrastructure)	0.1091	−0.0098t − 0.0214	-	-
Scenario 1 (infiltration storage facilities)	0.3621	−0.0072t − 0.0142	0.0012t − 0.0147	44
Scenario 2 (porous pavement)	0.2237	−0.0086t − 0.0188	0.0012t − 0.0136	48

## Data Availability

Data sharing not applicable.

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
