# Peer review of "Nature-Based Restoration Simulation for Disaster-Prone Coastal Area Using Green Infrastructure Effect"

_ijerph, 2023, doi:10.3390/ijerph20043096_

Round 1

Reviewer 1 Report (Previous Reviewer 3)

The idea and research on green infrastructure are very popular and needed, especially in the face of climate change. Water management is a problem not only in Korea but also in other countries. I appreciate all changes introduced by the Authors. However, many issues remained unsolved. The experiment design is very chaotic. The basic idea was good, but the form is questionable. I am sorry to say this but, in my opinion, paper should be rejected. I encourage you to revise the paper because your work is significant, however, you have to be more careful while planning experiment and describing it.

General comments:

1.      I suggest a little ‘polishing’ of the title. Maybe something like: ‘Nature-based restoration planning for disaster-prone coastal areas – green infrastructure effects’. Also, you should reconsider if you want to focus on the planning part.

2.      Application of porous pavements in public, private, industrial, transportation areas caused an increase of runoff. All your results show that PP has a negative effect. Therefore, maybe you should calculate runoff amount GI10%, GI20% and GI30% (as in fig. 1) without porous pavements? You may obtain better results.

3.      I do not understand how did you chose application rates for each area: in the public areas it was 10% (I suppose, it is not specified clearly), in the private areas it was 30%, in the industrial areas it was 15% or 30% (again, it is unclear). Why? How can you compare these results? Use one rate for every area type or show some justification of your choices. Also, in some cases basic scenario means no GI was applied, sometimes it is 15% or 30%. You should unify it throughout the paper.

4.      As in the previous version of your paper, you did not explain why porous pavements has a negative effect on runoff (increase).

5.      You did not show any specific restoration plan. Yes, you said that GI should be applied to the maximum but it is a very general and vague plan. Anyone could say that. You should be more specific and do not ignore the results showing that PP have negative effects. I suggest focusing more on your results and omitting the planning part. Or prepare it better.

6.      The discussion section is better but still very general, the same information is repeated throughout the text (using GI is positive, it should be applied as much as possible), it is lacking discussion of your results. What do they mean? What do they show?

Specific comments (line numbers are from the file with marked changes):

7.      Line 16: Did you mean ‘disaster-prone coastal areas’?

8.      Line 19: After 6 hours of what? You have to be more specific in the abstract section – the abstract should show what the article is about and encourage a potential Reader to read the rest of the paper.  Therefore, I think you should include a short description of your experiment design, now it is unclear what you have studied.

9.      Line 24: I suggest changing ‘plan’ to ‘tool’.

10.   I appreciate adding the study site description. It shows why you have chosen this area and why it is flood sensitive. One minor suggestion: please add information that Haeundae-gu is Busan district; line 92: ‘In this study, Haeundae-gu, a district of Busan, was selected…’.

11.   Line 104: please change the form of citation, I suggest : ‘using the data from the study by Song et al. [25]’

12.   Lie 107: If ‘the previous study’ refers to study by Song et al. [25] you should state it clearly, so ‘the previous study [25] was also utilized…’.

13.   Line 117: Total areas of what? Of district? Or of areas where green infrastructure could be applied?

14.   Fig. 1: Please add in the figure description or within the graphic meaning of the abbreviation ‘GI’. I know it is easy to guess but you have not introduced this abbreviation in the text.

15.   Figure 2: What blue, red and thick black lines mean? Are blue and red lines scenarios #2 ad #3 respectively? If so, then where is scenario #1?

16.    Figure 3 and Table 2 show the same data, please keep only one of them. The same with fig. 4 and tab. 3; fig. 5 and tab. 4; fig. 6 and tab. 5; fig. 7 and tab. 6.

17.   Line 207: What previous analysis? Please cite it here.

18.   In line 218-219 you say that three green infrastructure types were applied at the same rate (basic scenario) – what rate? What was the ratio of implementing GR (green roofs),  ISF and PP? 10, 20 or 30%?

19.   Lines 230-234: I’m sorry but I do not understand what do you man here “The scenario in which only the porous pavement is applied has higher runoff than the basic scenario in which the green infrastructure is applied in the same ratio. This is because in the basic scenario, 10% of green roof, 10% of infiltration storage facility, and 10% of porous pavement were applied within the biotope area ratio of 30%.” So this 30% consists of 10% of GR, 10% of ISF and 10% of PP? Also, I think it is not a sufficient explanation of this result. It is very interesting and should be further explained as a part of the discussion.

20.   Figure 4: Please explain the abbreviations in the fig. description (GR, ISF, PP).

21.   Lines 238-240: I this form this sentence means that only 9.4% of the GR and 5.5% of the ISF  of all GR and ISF reduced the runoff. Didn’t you mean that application of GR reduced the runoff by 9.4%?

22.   Lies 243—245: Accounted for what? Runoff reduction? Runoff increase?

23.   Lines 252-254: So 15% was applied for the basic scenario? And scenarios #1 and #2 was 30%. Why? Why did you use different ratios for basic scenario and variants? You cannot compare these results. Same with transportation area.

24.   Figures 4-7: A basic scenario should be a scenario without implementation of the GI at all. Then you could compare various scenarios, for example application of green roofs at a ratio of 10% etc.

25.   Lines 278-399: You do not have to repeat each time that in the case of public and private areas you applied green roofs, infiltration storage facilities and porous pavements, while for transportation and industrial areas only ISF and PP. You explained this in lines 155-158.

26.   Line 331: ‘the application rate differed in each area’ – why? How did you choose these rates?

27.   Lines 358-359: So what rates did you apply here?

28.   Lines 369-370 and 387-388: What was the rate of application?

29.   Line 432: What previous results?

30.   Lin 437: ‘porous pavement should be applied to other artificial ground areas’ – what areas? According to your results it showed only negative effects.

31.   Line 451: “according to the biotope area ratio in Korea” – what is this ratio? Is it forbidden to introduce more than 30% of GI?

32.   Line 474: The country of Korea suggested you these ratios? Please reshape this sentence.

33.   Line 478: “Porous pavement showed the lowest effect” – more accurate sentence would be “Porous pavement showed a negative effect”.

34.   Line 481: “Third, based on these results, nature-based restoration planning was presented” – it was very general plan.

35.   Line 489: More concrete and valid than what?

Author Response

I am really appreciate for giving me a detailed opinion even though my manuscript is insufficient. We revised the ovall manuscript as much as possible to reflect your comments.  I am attaching the file in response to your comments.

Reviewer 2 Report (New Reviewer)

In the introduction, I suggest to refer to sociohydrology. The main research problem of sociohydrology is human interaction with the environment in the context of flooding.

Also take into account the damage caused by gray infrastructure to the natural environment.

I'm not a native speaker, but the style should be improved in some places (e.g. line 96 - "In this study, a study was...").

Be sure to prepare a drawing showing the research area. In addition to the studied region, I suggest marking typhoon routes (e.g. as here https://doi.org/10.7837/kosomes.2020.26.7.881)

Lines 120-123. This sentence suggests that salt marshes and others are just as green ifrastructures as green roofs. I suggest you expand this statement and separate the "natural" green infrastructures from the "anthropogenic" one.

In figure 2, add a legend describing the red and blue lines.

In table 2, it is enough to write numbers with one digit after the dot.

Does figure 3 represent runoff or rainfall?

Line 201-202. This sentence is obvious. If you used the 40% ratio, it would give the best result.

The discussion covers the correct topics, but many of them have no reference to the literature.

Author Response

I am really appreciate for giving me a detailed opinion even though my manuscript is insufficient. We revised the ovall manuscript as much as possible to reflect your comments.  I am attaching the file in response to your comments.

Round 2

Reviewer 1 Report (Previous Reviewer 3)

Thank you for making the changes I suggested. The data has been sorted out, the discussion has been greatly enriched. I congratulate the Authors for their persistence and strength, because they have come a long way in improving the article – especially that they had to recalculate the models. The paper is very good in this form. I hope that it will indeed be implemented in nature-based restoration planning. I recommend publishing it in IJERPH.

One minor comment: I suggest adding a scale to figure 1.

This manuscript is a resubmission of an earlier submission. The following is a list of the peer review reports and author responses from that submission.

Round 1

Reviewer 1 Report

This study aims to establish a social-ecological restoration planning process for flood resilience in coastal regions through a qualitative approach based on system thinking and a quantitative approach through simulation modeling. The authors developed and apply a socioecological restoration planning process in Haeundae-gu, Busan. Such studies are of great significance and could provide a proposed plan to effectively respond in coastal regions for better disaster management techniques. I would like to recommend the publication of this work after the following minor revision and clarification:

·         The manuscript is not structured properly. It seems like a report. The authors are advised to replace the 2 and 3rd sections with Data & methodology and  Results & discussion respectively.

·         Please correct the citations and references style throughout the manuscript as per the MDPI format.

·          Out of four events, why was only Typhoon Chaba 2016 considered a case study?

·         Table 1: Please specify the duration of precipitation frequency.

·         To reduce the flood damages, the authors may also yield the maximum net benefit by considering expected annual damages and the annual equivalent cost of implementing, maintaining, repairing, replacing and rehabilitating the proposed flood resilience optimal plan in coastal areas.

Reviewer 2 Report

General Comments

The abstract need to be revised to briefly define the approach and methodology used in the research. The numerical results should also be presented (if any). By reading the abstract, I notice that a substantial improvement in English language and style is needed throughout the manuscript.

The aim and objectives of the study is broad and not clearly and specifically define. It is advisable for the authors to provide the procedure to briefly describe the step-by-step methodology used in the manuscript. For example, in Figure 4, I have no idea, how did you suddenly get the runoff simulation here. It is confusing. At this point, I will reject the manuscript. My other specific comments are provided as below.  

Specific comments

Tile

·       The title is too broad. It can be improved, and be more specific on the case study and method use.

Abstract

·       Line 12-14: Please revised the sentence.

·       Line 15-18: Pleased fine the approach used clearly in the abstract

·       Line 21-22: “Considering green infrastructure application by type…”. Do you mean “Comparing the type of green infrastructure being applied,…”

·       There is too many English language style that need to be improve throughout the manuscript.

·       Table 1: What simulation model are you using? GIS? HEC-HMS? This is confusing. What is the purpose of this parameter? This parameter will be use as input for what model?  

Reviewer 3 Report

Comments are in the file attached.
